# Anomalous $\delta^{15}$N values in the Neoarchean associated with an abundant supply of hydrothermal ammonium

Ashley N. Martin [1,2] ✉, Eva E. Stüeken [3], Michelle M. Gehringer [4], Monika Markowska[2,5], Hubert Vonhof [5], Stefan Weyer[1] & Axel Hofmann [6]

Unusually high $\delta^{15}$N values in the Neoarchean sedimentary record in the time period from 2.8 to 2.6 Ga, termed the Nitrogen Isotope Event (NIE), might be explained by aerobic N cycling prior to the Great Oxidation Event (GOE). Here we report strongly positive $\delta^{15}$N values up to +42.5 ‰ in ~2.75 – 2.73 Ga shallow-marine carbonates from Zimbabwe. As the corresponding deeper-marine shales exhibit negative $\delta^{15}$N values that are explained by partial biological uptake from a large ammonium reservoir, we interpret our data to have resulted from hydrothermal upwelling of $^{15}$N-rich ammonium into shallow, partially oxic waters, consistent with uranium isotope variations. This work shows that anomalous N isotope signatures at the onset of the NIE temporally correlate with extensive volcanic and hydrothermal activity both locally and globally, which may have stimulated primary production and spurred biological innovation in the lead-up to the GOE.

The remarkable variation of nitrogen (N) isotope values ($\delta^{15}$N = [($^{15}$N/$^{14}$N$_{sample}$)/($^{15}$N/$^{14}$N$_{air}$) − 1] × 1000) in Neoarchean sedimentary rocks, from −11‰[1] up to 50‰[2,3], hints at fundamental shifts in global marine N cycling prior to the initial rise of atmospheric oxygen during the great oxidation event (GOE). The time interval that features this particular cluster of strongly positive N isotope values around 2.8 to 2.6 billion years ago (Ga) has recently been termed the nitrogen isotope event (NIE) and suggested to relate to the advent of aerobic ammonium oxidation globally[4]. Previously, such occurrences have been explained in terms of either partial nitrification coupled to denitrification in a marine oxygen oasis[3] or NH$_3$ volatilisation under high-pH conditions in a lacustrine setting[2], whereby both scenarios invoke the loss of isotopically light N to the atmosphere. However, the operation of a limited aerobic N cycle during the Neoarchean is at odds with much of sedimentary $\delta^{15}$N record with an average value closer to 0‰, suggesting that the Archaean N cycle was dominated by biological N fixation utilising the molybdenum-iron nitrogenase co-factor from

3.2 Ga onwards[5] with limited evidence for aerobic N cycling and the presence of nitrate until the GOE[6,7]. A largely anoxic Archaean world is consistent with evidence from geochemical redox proxies that indicate only transiently elevated oxygen levels in shallow marine environments[8–11]. The NIE model is also difficult to reconcile with highly negative $\delta^{15}$N values in deep-water shales from the ca. 2.75 Ga Manjeri Formation, Zimbabwe craton, which are explained in terms of partial N assimilation into biomass from a deep, ammonium-rich reservoir[1]. This explanation requires an isotopically heavy N sink that has thus far remained elusive, rendering the isotopic mass balance and our understanding of the early N cycle prior to the GOE incomplete.

The Manjeri Fm (Ngezi Group, Belingwe greenstone belt) is a relatively thin (ca. 100 m) sedimentary succession that was deposited rapidly during a marine transgression of the Zimbabwe proto-craton due to crustal extension and subsidence, which was immediately followed by extensive submarine mafic and ultramafic volcanism

[1]Institute of Earth System Sciences, Leibniz University Hannover, 30167 Hanover, Germany. [2]Department of Geography and Environmental Science, Northumbria University, Newcastle upon Tyne NE1 8ST, UK. [3]School of Earth & Environmental Sciences, University of St Andrews, St Andrews KY16 9TS, UK. [4]Department of Microbiology, University of Kaiserslautern-Landau (RPTU), 67663 Kaiserslautern, Germany. [5]Max Planck Institut für Chemie, 55128 Mainz, Germany. [6]Department of Geology, University of Johannesburg, APK Campus, Auckland Park, PO Box 524, 2006 Johannesburg, South Africa. ✉e-mail: ashley.n.martin@northumbria.ac.uk

associated with the overlying Reliance and Zeederbergs formations at 2.746 ± 0.004 Ga[12–14] (Fig. 1). This period correlates with globally enhanced rates of mafic-ultramafic volcanic activity at ~2.75 Ga[14,15] associated with convective mantle overturning[16,17]. The sedimentary Cheshire Formation was deposited following this volcanic episode, which is temporally constrained by the eruption of the Ngezi volcanics at 2.75 Ga and the crystallisation of a dolerite intrusion at ~2.71 Ga[14]. Although the Manjeri and Cheshire Fm carbonates exhibit positive Eu anomalies (Eu/Eu* = (Eu/0.67Sm + 0.33Tb); defined as Eu/Eu* > 1) up to 6.1, this is regarded as a primary feature of the local ambient seawater rather than reflecting close proximity to a hydrothermal vent[18]. Positive Eu anomalies are a common feature of detrital-free Archaean chemical sedimentary rocks and can be attributed to a Eu excess under reducing conditions in Neoarchean seawater due to increased hydrothermal activity globally[19]. Moreover, the Manjeri and Cheshire Fm carbonates exhibit $^{87}Sr/^{86}Sr$ ratios as low as 0.70155[18], which is similar to that of the depleted mantle value at the time of deposition and indicates that carbonate sedimentation around the Zimbabwe protocraton occurred co-eval to abundant submarine greenstone volcanism and prior to the stabilisation of most Archaean cratons[18].

Here, we investigate shallow marine limestones from the ~2.75 Ga Huntsman Quarry (Bulawayo greenstone belt), which is stratigraphically correlated with the Manjeri Fm[18], and younger shallow marine limestones from the ~2.73 Ga Cheshire Fm. We focus on well-preserved carbonates with limited metamorphic alteration (lower greenschist facies metamorphism), which were screened according to their previously reported[18] stable carbon (C) and oxygen (O) isotope compositions ($\delta^{13}C$ and $\delta^{18}O$), and radiogenic strontium isotope ($^{87}Sr/^{86}Sr$) ratios (Table 1). We combine these existing data with measurements of bulk N and organic C isotopes ($\delta^{15}N_{bulk}$ and $\delta^{13}C_{org}$), and also analyse uranium (U) isotopes to assess the redox conditions of the depositional environment.

## Results

### Bulk nitrogen and organic carbon isotope data

The $\delta^{15}N_{bulk}$ values of the Cheshire and Manjeri formation carbonates are strongly positive, ranging from +26.0‰ to +35.0‰ (average = +31.1 ± 2.5‰; $1\sigma$; $n = 12$) and +28.1‰ to +42.5‰ (average = +37.5 ± 6.5‰; $1\sigma$; $n = 4$), respectively, with $\delta^{13}C_{org}$ values ranging from −39.1‰ to −29.7‰ (Fig. 2a; Table 2). We infer a predominantly organic origin for N in our samples due to the correlation between TN and TOC contents for all samples ($R^2 = 0.99$), which is also found when only considering the Manjeri Fm ($R^2 = 0.99$; Fig. 2b). Although a weaker relationship is found for the Cheshire Fm ($R^2 = 0.37$), this is likely because the latter are generally more silicified than the Manjeri Fm[18]. Furthermore, there is an outlier (sample BD49-E-C5), which exhibits anomalously higher TOC contents (2.28%) compared to the average value for the Cheshire Fm (1.00 wt%); excluding this sample results in a stronger relationship ($R^2 = 0.59$). The C/N ratios range from 68 to 231 mol/mol for the Cheshire Fm and 71 to 82 mol/mol for the Manjeri Fm (Fig. 2c). This range overlaps with that measured in the Serra Sul Fm, which also exhibits similarly positive N isotope values[4], but it is somewhat lower than the Tumbiana Fm samples, which exhibit very elevated C/N values up to 589 mol/mol[2].

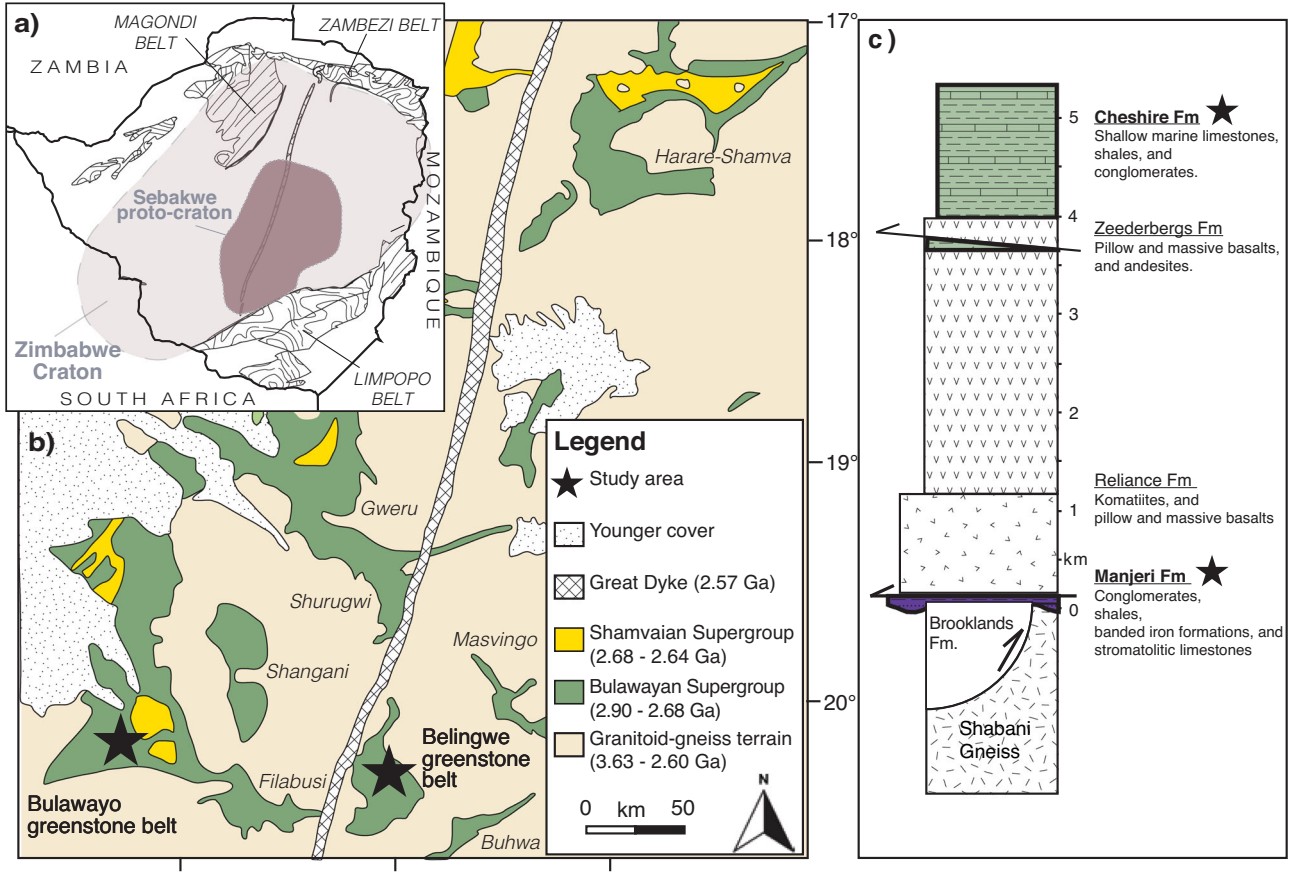

**Fig. 1 | Geological context of the Manjeri and Cheshire formations. a** Regional setting of the Zimbabwe craton. **b** Simplified geological map showing the Archaean granitoid (yellow shaded areas) and greenstone terrain (green shaded areas) of the Zimbabwe craton. **c** Generalised stratigraphy of the Bulawayan Supergroup of the Belingwe greenstone belt. Figure modified from ref. 61.

**Table 1 | Summary of sample descriptions and geochemical data for the Manjeri Fm and Cheshire Fm carbonates**

| Sample | Unit | Description | CaO (wt%) | MgO (wt%) | $\delta^{13}C$ (‰) | $\delta^{18}O$ (‰) | $^{87}Sr/^{86}Sr$ | Y/Ho | Ce/Ce* | Eu/Eu* | U (ppb) | Th (ppb) | Th/U | $\delta^{238}U$ (‰) | $\delta^{234}U$ (‰) |
|---|---|---|---|---|---|---|---|---|---|---|---|---|---|---|---|
| ZO4-17A | Manjeri Fm | Stromatolitic limestone | 54.1 | 0.9 | 0.53 | −20.32 | 0.701613 | 104.5 | 0.62 | 2.96 | 95 | 6 | 0.1 | −0.28±0.02 | 57.5±0.4 |
| ZO4-17B | | Stromatolitic limestone | 55.5 | 0.1 | 1.96 | −16.88 | 0.701571 | 156.5 | 0.48 | 6.10 | 53 | <1 | <0.1 | −0.31±0.07 | 297.3±12.8 |
| ZO4-17C | | Stromatolitic limestone | n/a | n/a | 0.56* | −20.52* | 0.701593 | 113.7 | 0.58 | 2.72 | 35 | 8 | 0.22 | −0.46±0.05 | 32.1±1.6 |
| ZO4-17D | | Stromatolitic limestone | 46.8 | 1.4 | −0.09 | −18.49 | 0.701877 | 86.5 | 0.66 | 1.84 | 157 | 18 | 0.11 | n/a | n/a |
| ZO4-17E | | Stromatolitic limestone | 53.0 | 0.8 | −0.30* | −15.27* | 0.701679 | 103.3 | 0.63 | 3.00 | 34 | <1 | <0.1 | −0.52±0.16 | 158.8±11.8 |
| ZO4-17F | | Stromatolitic limestone | 54.2 | 0.6 | −0.54 | −16.02 | 0.703631 | 60.1 | 0.90 | 1.73 | 880 | 33 | <0.1 | n/a | n/a |
| ZO4-27 | | Sheared limestone | 50.6 | 0.4 | 0.62 | −10.43 | 0.703398 | 84.1 | 0.66 | 4.37 | 19 | <1 | <0.1 | −0.43±0.03 | 8.5±0.9 |
| BD19 | Cheshire Fm | Limestone | 28.1 | 2.9 | −1.11* | −13.73* | 0.710222 | 24.0 | 0.93 | 1.26 | 218 | 202 | 0.93 | −0.10±0.15 | 241.5±10.4 |
| BD33 | | Limestone | n/a | n/a | −0.61* | −13.35* | n/a | n/a | n/a | n/a | n/a | n/a | n/a | −0.22±0.07 | 200.8±7.1 |
| BD37 | | Limestone | 38.0 | 2.2 | −1.97 | −16.26 | 0.712371 | 28.5 | 0.89 | 2.29 | 33 | 96 | 2.9 | −0.26±0.07 | 243.5±2.7 |
| BD43 | | Limestone | n/a | n/a | −1.24 | −16.25 | n/a | n/a | n/a | n/a | n/a | n/a | n/a | −0.32±0.03 | 130.1±2.0 |
| BD46 | | Limestone | n/a | n/a | −0.47* | −13.89* | n/a | 63.1 | 0.75 | 2.42 | 13 | 16 | 1.2 | −0.56±0.16 | 134.0±6.3 |
| BD49 | | Limestone | n/a | n/a | −1.63* | −15.02* | n/a | 37.5 | 0.89 | 1.80 | 89 | 190 | 2.1 | −0.41±0.01 | 203.9±4.6 |
| BD50 | | Limestone | 46.8 | 0.3 | 0.06 | −13.85 | n/a | 29.4 | 0.97 | 1.61 | 20 | 198 | 10.1 | n/a | n/a |
| BD51 | | Limestone | 44.5 | 0.7 | −0.11 | −15.20 | 0.712061 | n/a | n/a | n/a | n/a | n/a | n/a | n/a | n/a |
| BD52 | | Stromatolitic limestone | n/a | n/a | −0.39* | −13.91* | n/a | n/a | n/a | n/a | n/a | n/a | n/a | n/a | n/a |

Major element, strontium (Sr) isotope and rare-earth element data sourced from Hofmann et al.[18]. Additional stable isotope data ($\delta^{13}C$ and $\delta^{18}O$) measured in this study are indicated by asterisks (*) whereby all other data are from Hofmann et al.[18]. Uncertainty values for $\delta^{238}U$ and $\delta^{234}U$ represent 2 standard error.

## Uranium isotope compositions

The uranium isotope composition ($^{238}U/^{235}U$, expressed as $\delta^{238}U$, see Eq. 1) of the Cheshire Fm and Manjeri Fm carbonates range from −0.56 ± 0.16‰ to −0.10 ± 0.15‰ (2 standard error; 2 s.e.) and $\delta^{234}U$ values (see Eq. 2) range from +8.5 ± 0.9‰ to +297.3 ± 12.8‰ (2 s.e.; Table 1). There is a weak correlation between $\delta^{238}U$ and the $^{234}U/^{238}U$ activity ratios (expressed relative to secular equilibrium as '$\delta^{234}U$', where $\delta^{234}U_{sec.eq.} = 0$ ‰) for the Cheshire Fm ($R^2 = 0.48$) and no correlation for the Manjeri Fm carbonates ($R^2 = 0.06$; Fig. 3a).

## Discussion

The $\delta^{15}N_{bulk}$ values of the Manjeri and Cheshire formations are strongly elevated in comparison to the mean value of ca. 0‰ in Archaean sedimentary rocks[5,7,20]. Although there is a positive relationship between $\delta^{15}N_{bulk}$ and C/N ratios for both the Manjeri and Cheshire formations (Fig. 2c) that suggests some N devolatilization related to regional metamorphism, this cannot explain the highly positive $\delta^{15}N$ values. This is because the metamorphic grade of our samples is low (lower greenschist facies) and the isotopic fractionation factor (ε) determined for greenschist-facies metamorphism is relatively small (1.5 ± 1‰)[21], which can only yield values up to ca. +10‰ for a starting composition of 0‰. Moreover, there is no evidence for proximal hydrothermal processes that may have resulted in large isotope fractionation factors associated with N release from minerals at ~300 °C[22–24]. Therefore, the primary N isotope signature of the Manjeri Fm and Cheshire Fm carbonates was likely in excess of +20‰, consistent with other examples of the NIE globally[2–4].

Our positive $\delta^{15}N_{bulk}$ values may be complementary to negative $\delta^{15}N$ values down to −11‰ reported from the deep-water shales in the Manjeri Fm[1], which have been explained in terms of partial ammonium assimilation from a large, deep-water ammonium reservoir. Assuming the median ε value of −14‰ for this metabolism would imply that ca. 70–90% of the dissolved ammonium pool was removed via biomass assimilation[25] (Fig. 4). A necessary outcome of this hypothesis proposed by Yang et al. is the generation of a residual ammonium pool in seawater with a positive N isotope composition ranging from +18‰ to +34‰. This predicted range agrees well with our carbonate $\delta^{15}N$ values. The highest $\delta^{15}N$ values may also be explained by partial ammonium oxidation, as recently suggested to explain the NIE[4], which implies the availability of free oxygen in the water column.

Similar to most Archaean carbonates[26], shallow marine carbonates from the Manjeri and Cheshire formations lack true negative, shale-normalised (SN) Ce anomalies (Ce/Ce*$_{SN}$ = Ce/(0.5La + 0.5Pr), defined as (Ce/Ce*$_{SN}$ < 1), but $\delta^{238}U$ values as low as −0.56‰ (Table 1) indicate subtle redox variations during their deposition. The lowest $\delta^{238}U$ are considered the most reliable (maximum) estimates for the seawater $\delta^{238}U$ value at the time of deposition because the heavy U isotope, $^{238}U$, is preferentially reduced during early sedimentary diagenesis, as demonstrated in modern Shark Bay stromatolites[27] and other modern shallow marine carbonates[28]. Therefore, only $\delta^{238}U$ values higher than the modern seawater value (−0.4‰) may reasonably be attributed to sedimentary diagenesis. As samples exhibit $^{234}U$–$^{238}U$ disequilibrium with $\delta^{234}U$ values up to ca. +300 ‰ (Table 1), this indicates some degree of U mobility within the past ca. 1.5 Ma. However, the weak correlation between $\delta^{238}U$ and $\delta^{234}U$ (Fig. 3a) indicates that post-depositional alteration by weathering fluids had only a limited effect on the $\delta^{238}U$ redox proxy. In any case, this correlation further supports the interpretation that samples with the lowest $\delta^{238}U$ likely provide the most reliable information regarding the primary sedimentary signature.

The reliability of the $\delta^{238}U$ redox proxy in ancient carbonates can be further examined by considering the rare-earth element + yttrium (REY) patterns. For instance, the Y and holmium (Ho) elemental ratio, which remains constant at the chondritic Y/Ho ratio of 26 to 28 during

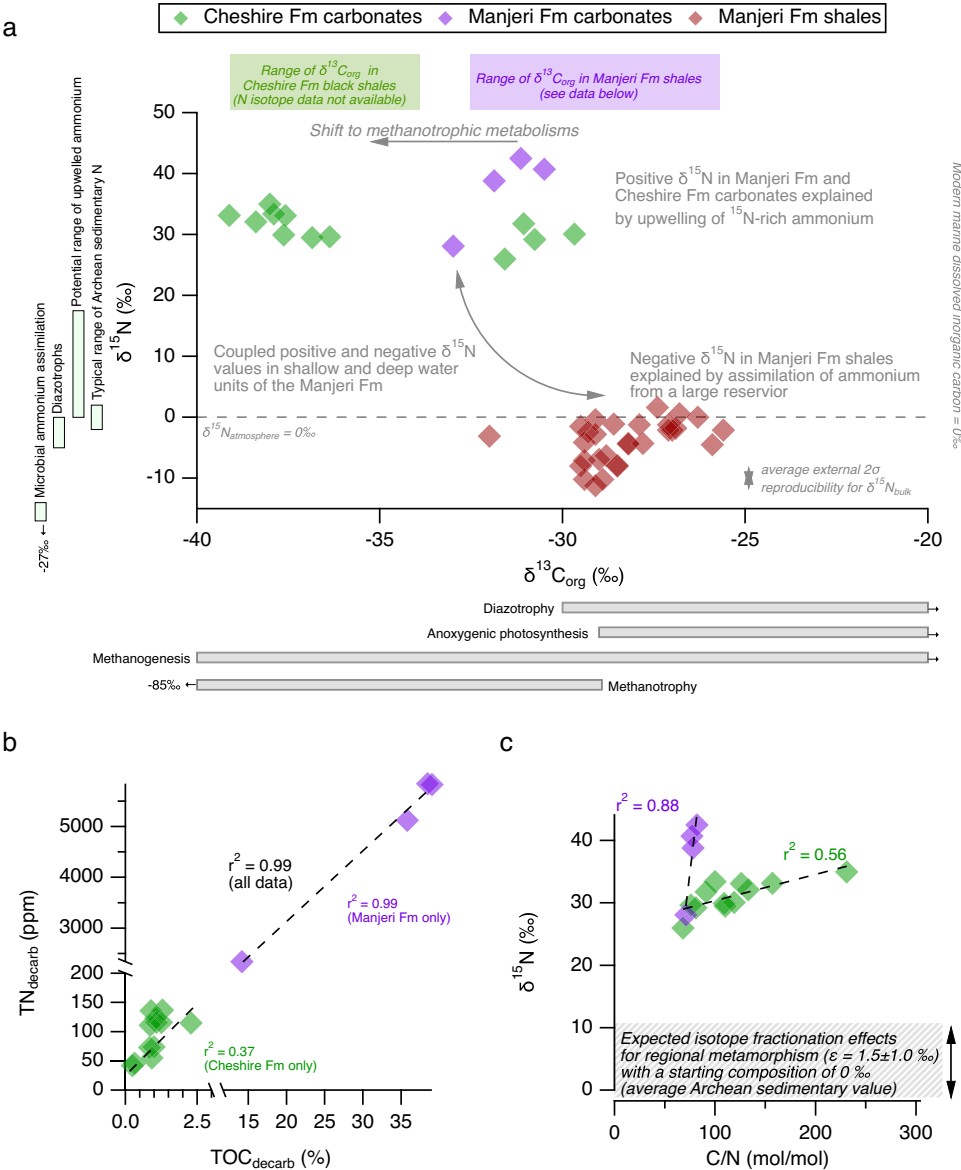

**Fig. 2 | Nitrogen ($\delta^{15}$N) and organic carbon isotope ($\delta^{13}$C$_{org}$) data for the Manjeri Fm and Cheshire Fm carbonates. a** Plot of $\delta^{15}$N vs $\delta^{13}$C$_{org}$ including data from Manjeri Fm shales[1] (red-filled diamonds) and Cheshire Fm shales[45] (green shaded area above plot), whereby grey-filled bars plotted outside the axes show the range of different metabolisms[62]. **b** Plot of TN vs TOC contents for the decarbonated residues. **c** Plot of $\delta^{15}$N vs C/N ratios with expected range shown for regional metamorphism effects[21]. Error bars represent 1$\sigma$ and those not shown are smaller than the marker symbol. Source data are provided as a Source Data file. Purple- and green-filled diamonds represent data from the Manjeri Fm and Cheshire Fm carbonates, respectively.

most geological processes but is fractionated in aqueous marine environments[29,30]. This results in modern seawater exhibiting a superchondritic Y/Ho ratio (>28) that is considered to represent a primary seawater signal in ancient carbonates[31,32]. Despite Y/Ho and U isotope representing two different chemical systems, they may record signals from the same source, i.e. a primary seawater signal, as both U isotope and REY signatures are typically well preserved during carbonate diagenesis[33]. A two-tailed *t*-test reveals that carbonates from the Manjeri and Cheshire formations with lower Y/Ho (<28) exhibit significantly higher $\delta^{238}$U values (mean ± 1 s.d. = −0.23 ± 0.09 ‰) than samples with Y/Ho greater than 37 ($p = 0.01$; mean ± 1 s.d. = −0.42 ± 0.12 ‰). This is consistent with a greater influence of detrital material for carbonates with higher $\delta^{238}$U and lower Y/Ho. According to the $\delta^{238}$U values of samples with a Y/Ho greater than 37, our data indicate that the local Neoarchean seawater that covered the Zimbabwe proto-craton possibly varied from a modern-like $\delta^{238}$U value of −0.4‰[34] to a

minimum of around -0.6‰ (Fig. 3b, c). Although these $\delta^{238}$U values overlap with the average value for modern open seawater (ca. −0.4‰[34]), we do not propose that the average oxidation state of the Neoarchean Ocean was similar to present. Lower $\delta^{238}$U values relative to modern seawater have also been interpreted to represent the onset of mildly oxidative weathering in other Precambrian sedimentary rocks[35,36], whereas in Phanerozoic sedimentary rocks, lower $\delta^{238}$U values are typically associated with a relative increase in the extent of seafloor anoxia and preferential reduction of $^{238}$U (refs. 37,38). We stress that the main significance of lower $\delta^{238}$U in Neoarchean carbonates is the implied presence of oxidised U$^{6+}$ in the water column. Mildly oxidising redox conditions may be reasonable if the local levels of dissolved oxygen in oxygen oases were related to the productivity of oxygen-producing cyanobacteria[39], which would be regulated by the supply of dissolved nutrients delivered from the upwelling of deep waters.

**Table 2 | Nitrogen and organic carbon isotope data for the Manjeri Fm and Cheshire Fm carbonates**

| Sample ID | Unit | TN$_{decarb}$ (ppm)[a] | ±1σ (ppm) | δ$^{15}$N$_{bulk}$ (‰) | ±1σ (‰) | TOC$_{decarb}$ (%)[a] | ±1σ (%) | δ$^{13}$C$_{org}$ (‰) | ±1σ (‰) | C/N (mol/mol) |
|---|---|---|---|---|---|---|---|---|---|---|
| BD49-E-C1 | Cheshire Fm | 137 | 10 | 29.5 | 0.5 | 1.29 | 0.12 | −36.85 | 0.13 | 110 |
| BD49-E-C2 | | 116 | 16 | 33.1 | 1.3 | 1.25 | 0.08 | −37.58 | 0.12 | 126 |
| BD49-E-C3 | | 74 | 3 | 33.2 | 0.6 | 0.99 | 0.03 | −39.12 | 0.05 | 157 |
| BD49-E-C4 | | 73 | 10 | 32.1 | 2.4 | 0.84 | 0.04 | −38.39 | 0.13 | 133 |
| BD49-E-C5 | | 115 | 8 | 35.0 | 0.5 | 2.28 | 0.21 | −38.01 | 0.02 | 231 |
| BD49-E-C6 | | 136 | 12 | 29.6 | 3.8 | 0.89 | 0.05 | −36.37 | 0.12 | 76 |
| BD49-E-C7 | | 116 | 12 | 29.9 | 3.5 | 1.08 | 0.06 | −37.63 | 0.06 | 109 |
| BD49-E-C8 | | 123 | 2 | 33.4 | 0.6 | 1.06 | 0.12 | −37.90 | 0.05 | 100 |
| BD52-B-C1 | | 42 | 2 | 26.0 | 2.0 | 0.24 | 0.01 | −31.58 | 0.07 | 68 |
| BD52-B-C2 | | 111 | 10 | 31.8 | 4.3 | 0.86 | 0.05 | −31.06 | 0.17 | 91 |
| BD52-B-C3 | | 46 | 5 | 29.2 | 3.2 | 0.32 | 0.00 | −30.76 | 0.02 | 81 |
| BD52-B-C4 | | 55 | n/a | 30.1 | n/a | 0.92 | 0.55 | −29.67 | 0.61 | 119 |
| ZO4-17C-G | Manjeri Fm | 5826 | 166 | 38.8 | <0.1 | 39.03 | 1.13 | −31.87 | 0.17 | 78 |
| ZO4-17C-E-2 | | 5121 | 47 | 42.5 | 0.2 | 35.79 | 2.36 | −31.14 | 0.02 | 82 |
| ZO4-17C-F-2 | | 2342 | n/a | 28.1 | n/a | 14.17 | 0.09 | −32.99 | 0.07 | 71 |
| ZO4-17C-D | | 5842 | 229 | 40.7 | 0.9 | 38.45 | 1.25 | −30.50 | 0.03 | 77 |

a: TN$_{decarb}$ and TOC$_{decarb}$ values represent the contents of the decarbonated residues.

The lack of true Ce anomalies despite variations in U isotopes may be reconciled by considering the redox potentials of these elements under aqueous conditions, whereby Ce$^{3+}$ is oxidised to Ce$^{4+}$ by oxygen at ca. +1 V at a circumneutral pH with low Ce$^{3+}$ concentrations found in seawater[40], whereas the two-step oxidation of U$^{4+}$ to U$^{6+}$ may occur at less oxidising conditions up to +0.3 V (ref. 41), which may constrain the upper limit of the redox potential in the shallow marine environment. Importantly, ammonium oxidation can occur at a redox potential of around +0.4 V under a circumneutral pH[42], which lies between the reduction potentials of Ce$^{4+}$/Ce$^{3+}$ and U$^{6+}$/U$^{4+}$. Thus, the strongly elevated δ$^{15}$N values suggest that the marine redox environment was at least transiently oxidising enough for ammonium oxidation to occur. This is plausible given that both modelling[43] and laboratory experiments[39] with cyanobacteria suggest that oxygenic photosynthesis can locally yield dissolved oxygen concentrations up to ~10 μM under Archaean conditions. However, molecular clock estimates suggest that most modern clades of ammonium oxidising bacteria and archaea emerged after the GOE in the Paleoproterozoic[44]. The low δ$^{13}$C$_{org}$ of the shallow-marine carbonates from Cheshire and Manjeri Fm (−39.1‰ to −29.7‰; Table 2) compared to the higher δ$^{13}$C$_{org}$ values in the deep-marine Manjeri Fm shales[1,45] (Fig. 2a) suggest that methanotrophy was restricted to shallow waters around the Zimbabwe proto-craton, possibly due to limited sulfate availability in deeper, more reducing waters. As there is a large isotope fractionation effect associated with methanotrophic ammonium oxidation to N$_2$O, this may yield elevated δ$^{15}$N values[46], especially in combination with the upwelling of a $^{15}$N-rich pool of residual ammonium due to biological ammonium assimilation in the deep basin[1].

Positive δ$^{15}$N$_{bulk}$ values coupled to low δ$^{13}$C$_{org}$ values may provide a signature of ammonium oxidation by methanotrophs under an abundant supply of methane in a hydrothermally influenced setting. Hydrothermal fluids are typically rich in ammonium (and methane) when circulating through sediment-covered oceanic ridges with modern hydrothermal fluids exhibiting elevated concentrations up to 16 mM[47–49]. In the Neoarchean, some hydrothermal vent fluids also had sufficiently high ammonium concentrations to facilitate the partial N utilisation by abiotic or biotic processes, such as those in the 2.7 Ga Abitibi basin[22]. Ammonium may also be sourced via the remineralisation of organic matter in marine sediments in the absence of hydrothermal fluid flow as, for instance, in the modern Black Sea. However, this process is inconsistent with our N isotope data because nutrient-rich reservoirs that accumulate due to intense basin stratification imply a limited upwelling of nutrients to the surface[50]. In contrast, buoyant hydrothermal vent plumes, which are typically warmer and less dense than the surrounding seawater, enable limited nutrients such as Fe to reach surface waters and stimulate primary productivity in the modern oceans on timescales of ~100 yr[51,52]. A similar scenario could therefore be plausible for hydrothermal ammonium in the Archaean. As ammonium may have been a limiting nutrient in the Neoarchean Ocean, it was probably rapidly scavenged from the water column, preferentially removing $^{14}$N via ammonium assimilation and archiving negative N isotope signature in the deep-water shales[1]. The remaining pool of $^{15}$N-rich hydrothermal ammonium could then have reached the shallow marine environment via upwelling and fuelled surface biological productivity in the form of microbial communities associated with stromatolites under weakly oxidising conditions due to oxygenic photosynthesis by cyanobacteria (Fig. 5).

Overall, the occurrence of strongly positive δ$^{15}$N in the ca. 2.75 Manjeri Fm and 2.73 Ga Cheshire Fm in the Zimbabwe craton further supports the global nature of the NIE, which includes the 2.68 Ga Serra Sul Fm (Amazonian craton)[4] and 2.72 Ga Tumbiana Fm (Pilbara craton)[2,3]. The link between elevated δ$^{15}$N values in shallow water carbonates and unusually negative δ$^{15}$N values in deep-water shales via ammonium assimilation may also explain both the extremely high values and greater variability of δ$^{15}$N values during the NIE. Furthermore, we show that the NIE began at least some 30 Ma earlier at around 2.75 Ga (Fig. 6), which temporally correlates with globally enhanced volcanism caused by increased magmatic production associated with mantle overturning[14,16,17]. Although global conditions during the Neoarchean were primed to supply enhanced fluxes of ammonium from hydrothermal vents in marine environments, not all places would have received enough ammonium to produce those extremely positive δ$^{15}$N values. However, distal hydrothermal fluxes of recycled ammonium were likely supplemented by key nutrients like methane and dissolved P (ref. 53) that could have triggered biological productivity during this time. In addition, enhanced fluxes of biologically useful transition metals such as copper, molybdenum and zinc[22,54] may have simultaneously catalysed novel biological diversification, thereby triggering the necessary conditions for the onset of the NIE and the expansion of life in the buildup to the GOE.

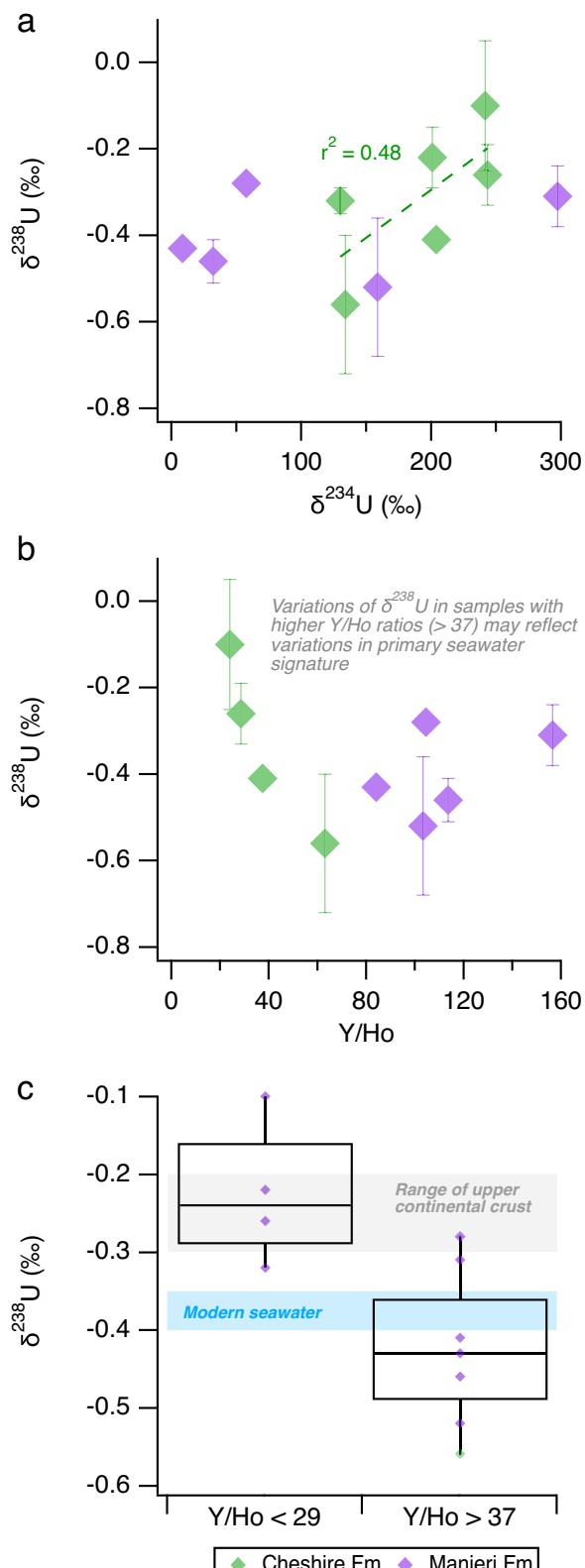

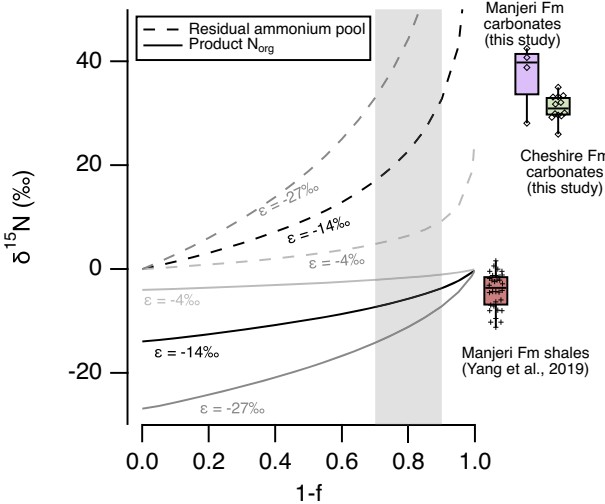

**Fig. 3 | Uranium isotope values and Y/Ho ratios[18] of the Cheshire Fm and Manjeri Fm carbonates. a** Plot of $\delta^{238}$U vs $\delta^{234}$U. **b** Plot of $\delta^{238}$U vs Y/Ho ratios. **c** box plots of samples with low (<29) and high (>37) Y/Ho ratios where the centre line shows the median, box limits show the upper and lower quartiles, and whiskers show 1.5 times the interquartile range. Error bars in panels **b** and **c** represent 2 standard error and those not shown are smaller than the marker symbol. Source data are provided as a Source Data file. Purple- and green-filled diamonds represent data from the Manjeri Fm and Cheshire Fm carbonates, respectively.

**Fig. 4 | The modelled residual ammonium pool following assimilation into the biomass in the deep basin and upwelling onto the shallow proto-cratonic shelf (modified from Yang et al.[1]).** Isotope fractionation factors represent experimentally derived values from Hoch et al.[25] and the grey shaded area represents the likely range for $1-f$ (following Yang et al.[1]). Box plots show the range of measured values in the various sedimentary facies where the centre line shows the median, box limits show the upper and lower quartiles, whiskers show 1.5 times the interquartile range. Source data are provided as a Source Data file.

oven, samples were washed three times with 18.2 MΩ/cm DI-H$_2$O to remove acidic residues. An appropriate amount of dry sample residue was then weighed into tin capsules and analysed with an elemental analyser for flash combustion (EA-IsoLink) coupled to a continuous-flow isotope-ratio mass spectrometer (MAT253 CF-IRMS) via a Conflo IV (all Thermo Fisher)[55]. Isotopic values were calibrated with the international reference materials USGS-40 and USGS-41 with USGS-62 as a secondary standard, which yielded average $\delta^{15}N_{bulk}$ and $\delta^{13}C_{org}$ values of $+20.26 \pm 0.18‰$ ($n = 17$; $1\sigma$) and $\delta^{13}C_{org} = -14.77 \pm 0.07‰$ ($n = 6$; $1\sigma$), respectively, which are consistent with previously reported values of $+20.17‰$ and $-14.79‰$, respectively. Devonian shale SDo-1 was also processed through the entire procedure and yielded average $\delta^{15}N_{bulk}$ and $\delta^{13}C_{org}$ values of $-0.44‰ \pm 0.24‰$ ($n = 4$; $1\sigma$) and $-30.16 \pm 0.29‰$ ($n = 4$; $1\sigma$), respectively. Fourteen carbonate samples were analysed in duplicate, yielding average reproducibilies of $\pm 1.70‰$ and $\pm 0.09‰$ for $\delta^{15}N_{bulk}$ and $\delta^{13}C_{org}$ ($1\sigma$), respectively. The TN and TOC contents of the decarbonated residues (TN$_{decarb}$ and TOC$_{decarb}$) were determined from peak areas of the IRMS analysis and calibrated with a series of USGS-41 measurements. Carbonate contents of each sample were estimated by weighing an aliquot of powder before and after treatment with 2 M HCl. Where we report total concentrations, decarbonated denotes the siliciclastic fraction remaining after treatment with HCl, as reported by Thomazo et al.[3]. Isotopic ratios are reported relative to atmospheric air for $\delta^{15}N_{bulk}$ and VPDB for $\delta^{13}C_{org}$.

Stable carbon and oxygen isotope values ($\delta^{13}$C and $\delta^{18}$O) of carbonate powders were analysed following methods previously

## Methods

### Stable isotope analyses

Nitrogen and carbon isotope values ($\delta^{15}N_{bulk}$ and $\delta^{13}C_{org}$) and their abundances of carbonate powders were analysed at the University of St Andrews (as previously described in ref. 2). Samples were first decarbonated by heating with 2 M HCl (reagent grade) at 70 °C overnight and centrifuged to remove the acid. Before drying in a closed

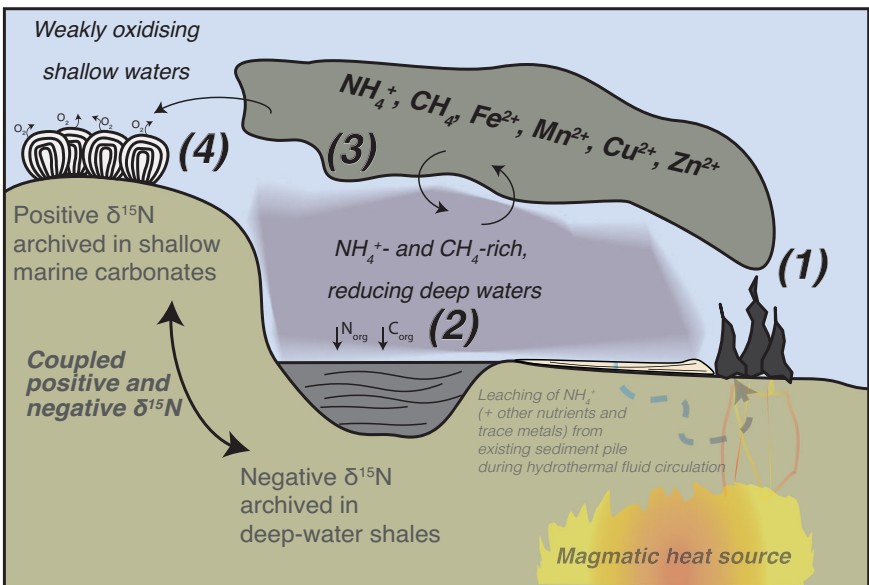

**Nitrogen cycle above Zimbabwe proto-craton at ca. 2.75 Ga**

1. Abundant supply of $NH_4^+$ and $CH_4$ from distal hydrothermal venting
2. Assimilation of $NH_4^+$ in deep-water shales producing a negative N isotope signature
3. Upwelling of buoyant, nutrient-rich hydrothermal plumes to surface waters
4. Strongly positive N isotope signature archived in stromatolitic carbonates
± partial oxidation of $NH_4^+$ by methanotrophs under $CH_4$-rich, weakly oxidising conditions

**Fig. 5 | Conceptual model of the submerged Zimbabwe proto-craton at ca.** 2.75 Ga to explain the coupled positive and negative nitrogen isotope values in terms of hydrothermal ammonium upwelling. Hydrothermal fluids rich in dissolved ammonium ($NH_4^+$) and other key nutrients are released in the deep basin and accumulate in the deeper waters, which are assimilated by biological organisms and produce negative nitrogen isotopes values in deep water sediments that eventually form shale rocks. The remaining dissolved $NH_4^+$ that reaches the surface due to upwelling processes is enriched in $^{15}N$ and archived in shallow-water carbonates.

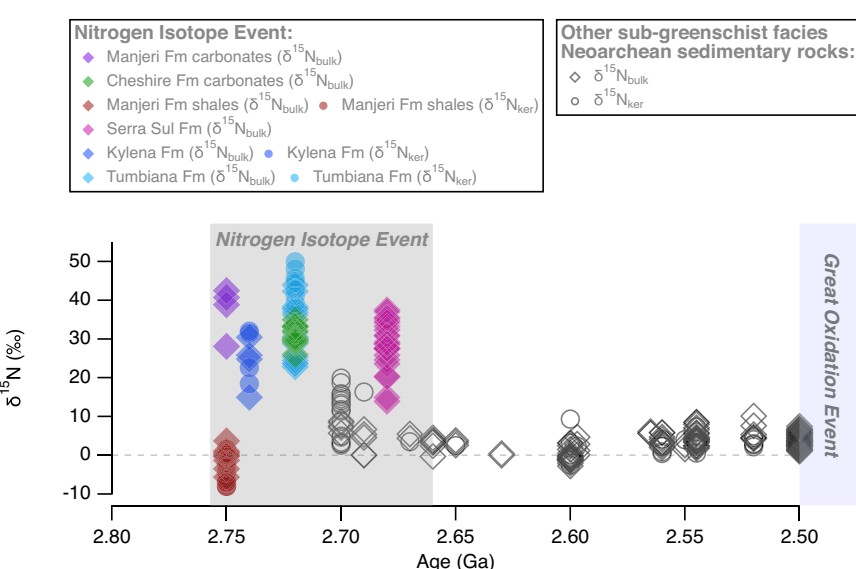

**Fig. 6 | Nitrogen isotope data ($\delta^{15}N$) for well-preserved (sub-greenschist to lower-greenschist facies) Neoarchean sedimentary rocks deposited between 2.80 and 2.45 Ga. Data were compiled by Stüeken et al.[63] and supplemented with additional data from the Manjeri Fm and Cheshire Fm carbonates (this study), Manjeri Fm shales[1] and the Serra Sul Fm[4]. Source data are provided as a Source Data file. Filled diamonds and circles represent $\delta^{15}N_{bulk}$ and $\delta^{15}N_{ker}$ data from various localities deposited during the Nitrogen Isotope Event, respectively, where purple: Manjeri Fm carbonates, green: Cheshire Fm carbonates, red: Manjeri Fm shales, pink: Serra Sul Fm, and blue: Tumbiana Fm and Kylena Fm.

described in Pallacks et al.[56] on a Thermo Delta V mass spectrometer equipped with a GASBENCH-II preparation device at the Max Planck Institute for Chemistry. Approximately ~20 to 50 μg of $CaCO_3$ sample was placed in a He-filled 12 ml exetainer vial and digested in water-free $H_3PO_4$ at a temperature of 70 °C. Subsequently, the $CO_2$–He gas mixture is transported to the GASBENCH in Helium carrier gas. In the GASBENCH, water vapour and various gaseous compounds are separated from the He-$CO_2$ mixture prior to sending it to the mass spectrometer. Isotope values are reported as $\delta^{13}C$ and $\delta^{18}O$ values relative to Vienna Pee Dee Belemnite (VPDB). A total of 20 replicates of two in-house $CaCO_3$ standards are analysed in each run of 55 samples. $CaCO_3$ standard weights are chosen so that they span the entire range of sample weights of the samples. After correction of isotope effects related to sample size, the reproducibility of these standards typically is better than 0.1‰ ($1\sigma$) for $\delta^{18}O$ and for $\delta^{13}C$.

### Uranium isotope measurements

Uranium isotope measurements were conducted following methods previously described in Martin et al.[27] and are briefly given here. Depending on sample availability and previously measured U concentrations, approximately ~300–1000 mg of stromatolite powder was leached with 20 mL 2 M HCl at room temperature for 24 h. The samples were centrifuged and the solutions were retained for analyses. Prior to column chromatography, the samples were evaporated at 80 °C to incipient dryness and a U double spike (IRMM-3636a)[57] was added to the samples, targeting a $^{236}U/^{235}U$ of ~3 and a molar U sample-spike ratio of ~20–25. To separate U from the carbonate matrix, column chromatography was conducted according to Weyer et al.[58] using the Eichrom UTEVA resin and 150–300 ng U was typically loaded. Following column chromatography, 0.1 mL $HNO_3$ (65%) and 0.1 mL $H_2O_2$ (30%) were added and evaporated at 80 °C to incipient dryness. The residue was then redissolved in 3% (v/v) $HNO_3$ to yield final solutions with U concentrations ranging from 50 to 100 ppb.

Isotopic measurements were conducted using a Thermo Scientific™ Neptune Plus™ in low-resolution mode with a Cetac Aridus 2 sample introduction system (dry plasma conditions) at LUH following Noordman et al.[38]. A standard Ni H sampler cone and X skimmer cone setup typically achieved >1 V/ppb sensitivity for $^{238}U$. The $^{233}U$, $^{235}U$ and $^{236}U$ isotopes were measured using Faraday detectors with $10^{11}$ $\Omega$ resistors and $^{238}U$ was measured with a $10^{10}$ $\Omega$ resistor whereby $^{234}U$ isotope was measured with a $10^{13}$ $\Omega$ resistor. The abundance sensitivity of $^{238}U$ on $^{236}U$ was monitored to ensure it was <1 ppm. Instrumental mass bias was corrected using the $^{233}U/^{236}U$ ratio according to the exponential law. Measurement sequences were performed using a standard-sample-bracketing method relative to a CRM-112A standard solution to calculate $\delta^{238}U$ (Eq. 1) and $\delta^{234}U$ are given according to Eq. 2 relative to the secular equilibrium (SE) of $^{234}U/^{238}U = 54.891 \pm 0.094 \times 10^6$ ($2\sigma$)[59]. Uranium isotope ratios are reported according to convention using delta notation (in ‰), given as:

$$\delta^{238}U = [(^{238}U/^{235}U)_{sample}/(^{238}U/^{235}U)_{CRM112A} - 1]^*1000 \quad (1)$$

$$\delta^{234}U = [(^{234}U/^{238}U)_{sample}/(^{234}U/^{238}U)_{s.e.} - 1]^*1000 \quad (2)$$

All $\delta$-values of samples represent triplicate measurements where uncertainty values represent $2\sigma$ standard error (2 s.e.) for both $\delta^{238}U$ and $\delta^{234}U$. Reference materials were measured throughout the measurement sequence to monitor the instrument performance and a limestone (JLs; Geological Survey of Japan) was also processed with each batch of samples for column chromatography. The average $\delta^{238}U$ values of IRMM-184, Reimep-18a and JLs were −1.17 ± 0.04‰ ($2\sigma$, $n = 9$), −0.25 ± 0.07‰ ($2\sigma$, $n = 9$), and −0.36 ± 0.08‰ ($2\sigma$, $n = 3$), respectively, and their average $\delta^{234}U$ values were −28.0 ± 1.0‰, 34.4 ± 2.3‰, and 34.3 ± 0.5‰, which are all consistent with reported values[60]. Total

procedure blanks from leaching and column chromatography were <4 ng and no blank corrections were applied to the data.

## Data availability
The stable isotope data generated in this study for nitrogen, organic carbon and uranium are provided in the Source Data file and uploaded in a Figshare repository (https://doi.org/10.6084/m9.figshare.27632010). Source data are provided with this paper.

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

## Acknowledgements

Funding for A.N.M. and S.W. (WE 2850/17-1), and M.M.G. (GE 2558/4-1) was provided by the German Research Foundation (DFG) priority programme SPP-1833 Building a Habitable Earth. A.N.M. was supported by additional funding from the DFG priority programme SPP-2238 Dynamics of Ore Metals Enrichment (MA 9571-3-1). E.E.S. acknowledges support from a UK Natural Environment Research Council (NERC) Frontiers grant (NE/V010824/1) and a Leverhulme Trust grant (RPG-2022-313). M.M. acknowledges funding from a Royal Society Award (URF \R1\231546). A.H. acknowledges logistical support from the Department of Geology, University of Zimbabwe.

## Author contributions

A.N.M. and E.E.S. performed the nitrogen isotope measurements. A.N.M. conducted the uranium isotope analyses. M.M. and H.V. obtained inorganic carbon and oxygen stable isotope data. A.N.M., M.M.G., E.E.S., S.W. & A.H. developed the concept and designed the experimental approach. A.N.M. wrote the initial draft. All authors contributed to reviewing and editing the paper at all stages.

## Competing interests

The authors declare no competing interests.

## Ethics

We affirm that all geological materials were collected in a responsible manner and in accordance with relevant permits and local laws. A local researcher (Axel Hofmann) collected the samples and is a co-author of this paper. Local and regional research relevant to this study has been cited where appropriate.
