## [Peer Review File · Nature Communications]

Anomalous $\delta^{15}\text{N}$ values in the Neoproterozoic associated with an abundant supply of hydrothermal ammonium

Corresponding Author: Dr Ashley Martin

Version 0:

Reviewer comments:

Reviewer #1

(Remarks to the Author)

This is a very interesting manuscript. In the manuscript, the authors present some high $\delta^{15}\text{N}$ values, as high as 40‰, acquired from shallow-water carbonate of the Cheshire Fm and the Manjeri Fm, about 2.75 Ga. In combination with the low $\delta^{15}\text{N}$ values reported from coeval deeper-water shales, the authors suggest that these high $\delta^{15}\text{N}$ values were caused by ^{15}N -rich ammonium, which is ascribed to incomplete ammonium assimilation during the upwelling from the deep ocean. In a step forward, the authors propose that the ammonium-rich deep water was attributed to the hydrothermal remobilization of sediment-bound ammonium.

The manuscript is well written, and the C and N isotopic data are solid. I generally agree with the authors' interpretation and have some comments for the authors to consider during revision.

(1) I think the bulk $\delta^{15}\text{N}$ can be well interpreted by the residual ammonium hypothesis. However, in order to interpret the NANOSIMS $\delta^{15}\text{N}$ data, the authors involve the partial ammonium oxidation hypothesis. It is possible. However, I think the evidence for the argument of oxic environment is not convincing. I am not an expert on the U isotope. The $\delta^{238}\text{U}$ data reported here is very similar to the modern seawater. Does it mean the ocean at 2.75 Ga had a redox condition similar to the contemporary ocean? Although the authors suggest that the original $\delta^{238}\text{U}$ has been preserved in the samples with high Y/Ho values. What is the logic here? They are two different chemical systems. I would like to see more evidence for this point.

(2) The authors compared the differences in the redox potentials between U and Ce. How about those between U and N? It would be better to add this to the main manuscript. In addition, was there any contribution from ammonium oxidation by iron oxides?

(3) The authors argue that the ammonium in deep water was sourced from hydrothermal fluids based on the positive Eu anomaly and low $^{87}\text{Sr}/^{86}\text{Sr}$ values. I agree that hydrothermal fluids were widespread during this time period. It seems there is a gap in the linkage between the hydrothermal fluids and ammonium. How do you exclude the potential source from organic matter remineralization? Such as the modern stratified basins.

(4) The implication for the GOE is not convincing. The tectonic backgrounds between these two periods might be substantially different. It seems the GOE was preceded by a tectonic quiescence.

Minor points:

Line 23: Is this the first in situ NanoSIMS $\delta^{15}\text{N}$ data report?

Line 54: change fractionations to variation range.

Line 113: -39.1‰? Double check.

Lines 142-144: Double-check the writing here.

Line 192-195: The molecular tree analysis generally has a large error bar.

Line 247: Delete " $\delta^{13}\text{C}_{\text{org}} =$ ".

Line 234 and 235: These two titles have to be changed.

Reviewer #2

(Remarks to the Author)

General comments

This paper would be timely to publish as it reports another new occurrence of highly enriched d15N values in sedimentary rocks from the 2.6-2.8 Ga time interval. Until very recently such values were so rare that it was not possible to see them as informative about processes shaping the earth system. A new occurrence of these values has just been reported in Pellerin et al 2024 (that to be honest I co-authored) who suggest that they are actually informing about paleo-environmental conditions at the global scale (i.e. crossing the redox threshold for ammonium oxidation), and proposed to name the time period within which these values occur, the Nitrogen Isotope Event (NIE). The present manuscript, by reporting another occurrence of such values comforts the Pellerin et al 2024 proposition of a global event (NIE) and my advice would be to use it to give more weight when making the point of a global process.

The present manuscript proposes another interpretation for these values: ammonium assimilation from a large ammonium reservoir, a hypothesis that has never been really supported so far for lack of convincing data. The comparison to extremely positive d15N data to other data available from contemporaneous sedimentary formation in the same basin (with less negative and in cases negative d15N values), provide support for the idea that they result from ammonium assimilation in a dominantly anoxic ocean, possibly accompanied by ammonium partial oxidation (The combination of both processes makes it easier to accommodate the extremely high d15N values). To validate the idea that ammonium could be stabilized on anoxic waters, the authors use Ce anomaly and U isotope ratio as redox tracers. They confirm anoxic conditions. Overall, this reasoning seems good to me and I think it should definitively be shared with the community.

From there, the authors take another step by proposing a process allowing this large ammonium reservoir, i.e. hydrothermalism associated with globally enhanced magmatism. To make this point they use evidence for hydrothermalism reported in other publications and spatial variations in the available d15N values. This is OK for me up to this point.

I am not convinced however by the upscaling or the hydrothermalism source of Nitrogen to the basing scale and beyond (as stated in the title). Following my reasoning, this would require a massive amount of ammonium released in many basins while ammonium release is not so frequent in hydrothermalism today, and an initial source of nitrogen which I to my knowledge can only be of sedimentary origin. The proposed process for the buildup of a large ammonium reservoir would thus require that the globally enhanced magmatism is associated with magma emplacement in organic -rich rocks, for which I do not think that we have evidence. I may be mistaken, and may-be they are already out there. In which case this part of the paper just needs additional work to emphasize and discuss them.

Another weak point in this paper, from my point of view, is the use of Nano-SIMS d15N results. Using this method, the authors determine the internal d15N variability of two samples and their bulk value (which is compatible with the direct bulk d15N measurement). They also use the whole d15N range to comfort their proposed hypothesis of ammonium assimilation. I find it tricky to use and discuss these results in such a short paper. In the present form the manuscript lacks detailed explanations and tests to validate the precision of the results. It is therefore not clear to me (non-specialist of Nano-SIMS analyses) if they are meaningful.

I think removing them from the paper or from the main text would make it easier to read it and would allow to go straight to the point without distracting the reader with data that are less convincing. On the other hand, I admit that such data are rare and need to be presented to the community at some stage. I would suggest to use them in another paper though.

If it is decided to maintain this part, I would need more information/clarification in the method section. Here are my questions:

- Why is a d13C value given for the synthetic N-doped SiC standard and not d15N? how was the 15N ratio determined in the standard? How precisely is it known? how does this precision translate into the d15N determination?

- As to the matrix effect, it is specified that it adds an additional uncertainty for the N-isotopic ratios of ~15‰? how is it determined in the present study? To which uncertainty are the 15‰ added? in other words, what is the total uncertainty of SIMS d15N measurements?

- Given that N/C are poorly reproducible (with a factor of 4), I would expect the d15N to be as well. Are there any relationship between d15N and C/N? that could point toward a stronger matrix effect than expected?

- would the recalculated bulk d15N from nano-SIMMS data be comparable to the bulk value for samples with a low d15N?

Figure 2: can you indicate what is the legend of the data plotted in the figure 2, C? is one symbol equivalent to one measurement or to the average of several measurement of one sample? What is the error bar indicating? According to the text it corresponds to several analyses of two samples. It would be good to add the recalculated bulk values in the figure.

Line 117 to 122: I do not understand the point that is being made here. And I have the feeling it is even counterproductive?

Line 143: add "using" before Cerium...

Hoping this helps

All the best

Version 1:

Reviewer comments:

Reviewer #1

(Remarks to the Author)

I am generally satisfied with the authors' responses to my comments and their revisions to the manuscript. I have added some further comments to the PDF document attached here for the authors' consideration.

Reviewer #2

(Remarks to the Author)

I think that the comments from the first round of reviews have been taken into account. The proposition of hydrothermalism ammonium supply large enough for allowing isotope expression of ammonium assimilation now seems totally convincing to me.

I confirm that at the time of this paper submission, Pellerin et al' paper had not been published. I apologize if I did not make this clear. My intention was not to make an accusation but to allow the paper to take this into account in order to go a step further.

I have only minor comments to make:

Line 17-18 please rephrase to make it apparent that the NIE does not correspond only to the positive d15N values but to the time period when they occur.

Line 33 similarly, rephrase as "the time interval that contains this particular cluster..."

Although in most places it is acknowledged that ammonium partial oxidation may contribute to the d15N signature, it seems to me that lines 130-133 and 183-185 may have been overlooked since they argue for the absence of molecular O₂, and hence for no or negligible ammonium oxidation.

Table 2: If the carbonate content was determined by the mass loss during acid attack as stated in the method section, it should be indicated in this table. This would then make it apparent that the TOC percent reported in Table 2 (which is by definition the organic content of the bulk sample) is in fact the TOC_{decab} (i.e. the TOC content of the decarbonated fraction). Please either change TOC to TOC_{decab} or calculate the TOC.

It might be necessary to change this in other places in the manuscript (such as the method section, figure 2b).

Finally, reading this second version, I wonder if it would be worth mentioning that this ammonium assimilation hypothesis also explains both the extremely high values and the extreme variability of d15N in the NIE. The Neoproterozoic might indeed be particularly prone to ammonium supply by hydrothermalism globally, but not all places would receive enough ammonium to produce those extremely positive d15N values. I find that this would give even more weight to the paper but this is just a suggestion.

All the best

Response to reviewers (NCOMMS-24-38840-T)

A point-by-point response document the changes to the manuscript corresponding to the reviewer comments are outlined in this document. Please find the reviewer comments (RC) below in *bold grey italics* and our author responses (AR) in plain black text. Where necessary, we have quoted the appropriate line numbers (LN) in the manuscript (MS) with track changes turned off.

Reviewer #1 (Remarks to the Author):

RC1 This is a very interesting manuscript. In the manuscript, the authors present some high d15N values, as high as 40‰, acquired from shallow-water carbonate of the Cheshire Fm and the Manjeri Fm, about 2.75 Ga. In combination with the low d15N values reported from coeval deeper-water shales, the authors suggest that these high d15N values were caused by 15N-rich ammonium, which is ascribed to incomplete ammonium assimilation during the upwelling from the deep ocean. In a step forward, the authors propose that the ammonium-rich deep water was attributed to the hydrothermal remobilization of sediment-bound ammonium. The manuscript is well written, and the C and N isotopic data are solid. I generally agree with the authors' interpretation and have some comments for the authors to consider during revision.

We thank the reviewer for their positive comments and address their specific comments in the following.

RC2 I think the bulk d15N can be well interpreted by the residual ammonium hypothesis. However, in order to interpret the NANOSIMS d15N data, the authors involve the partial ammonium oxidation hypothesis. It is possible. However, I think the evidence for the argument of oxic environment is not convincing.

Regarding the NanoSIMS data – these are now removed from the revised MS due to comments from both reviewers and particularly Reviewer 2 who suggested that a dedicated, longer format was required to properly evaluate this data and as such, these results will now go into a different publication. However, regarding the partial ammonium oxidation hypothesis, Reviewer 2 actually proposed that we should further consider and discuss this more in our manuscript, especially with respect to the recent publication by Pellertin et al., 2024 in Nature that suggests similarly positive N isotope values represent the onset of aerobic N cycling globally. This directly contradicts the opinion of Reviewer 1, which reflects the debate in the community regarding redox conditions before the GOE and, for instance, the potential existence of oxygen oases. Therefore, we put great effort in measuring multiple redox proxies (Ce and U isotopes) in our study. Overall, these data from Ce and U isotopes agree more with the view of Reviewer 1 that the redox environment was predominantly anoxic. This is now clearly stated in the revised manuscript in LN179-184:

“Methanotrophic ammonium oxidation to N₂O yields a large isotope fractionation effect and would result a ¹⁵N-rich pool of residual ammonium⁴⁴. However, this process requires oxygen as an electron acceptor and this contradicts the lack of Ce anomalies

in our samples, suggesting that the redox potential of the local environment was lower. On this basis, we prefer upwelling as the primary mechanism to explain ¹⁵N enrichments in our study area.”

RC3 I am not an expert on the U isotope. The d238U data reported here is very similar to the modern seawater. Does it mean the ocean at 2.75 Ga had a redox condition similar to the contemporary ocean? Although the authors suggest that the original d238U has been preserved in the samples with high Y/Ho values. What is the logic here? They are two different chemical systems. I would like to see more evidence for this point.

Regarding the Y/Ho and U isotope data, although these are two different chemical systems, they may record signals from the same source, i.e. a primary seawater signal, and this is what we were suggesting. This link was not perhaps unclear to readers who are not experts in these systems and to address this, this point has now been made clearer in the revised MS (LN142-147):

“The reliability of the $\delta^{238}\text{U}$ redox proxy in ancient carbonates can be further examined by considering yttrium (Y) and holmium (Ho) elemental ratios, which remains constant at the chondritic Y/Ho ratio of 26 to 28 during most geological processes but is fractionated in aqueous marine environments^{31,32}. This results in modern seawater exhibiting a superchondritic Y/Ho ratio (>28) that is considered to represent a primary seawater signal in ancient carbonates^{33,34}.”

RC4 The authors compared the differences in the redox potentials between U and Ce. How about those between U and N? It would be better to add this to the main manuscript. In addition, was there any contribution from ammonium oxidation by iron oxides?

In the original MS, we did not include the redox potential of N species because our discussion of Ce and U was limited to inorganic redox reactions, whereas most N-based reactions are biological and we felt it is more logical to discuss these in terms of available electron acceptors. However, we added additional text to make this point clearer, which also relates to a previous comment by the reviewer regarding local redox conditions (RC2). This rewritten revised MS.

Regarding ammonium oxidation by iron oxides, we assume that this relates to the study by Pellerin et al. (2023; Geobiology), which analysed cherts and BIFs from the ca. 3.4 Ga section of the BARB3 drill core. We do not feel that our data from shallow water carbonates is comparable and do not discuss this.

RC5 The authors argue that the ammonium in deep water was sourced from hydrothermal fluids based on the positive Eu anomaly and low ⁸⁷Sr/⁸⁶Sr values. I agree that hydrothermal fluids were widespread during this time period. It seems there is a gap in the linkage between the hydrothermal fluids and ammonium. How do you exclude the potential source from organic matter remineralization? Such as the modern stratified basins.

The point we were trying to make is that hydrothermal fluids accelerate organic matter remineralisation but we now realise that this was not sufficiently clear in the original MS. To support our point, we now state that modern hydrothermal fluids are rich in ammonium if they are covered by sediments. And we also cite work supporting the claim that Archean hydrothermal vents were also rich in ammonium. We also discuss the reviewer's point regarding stratified basins and stress that because they're stratified and so the nutrients such as ammonium do not reach the surface waters. However, this discussion point is important to include and is now discussed. All these points are addressed in LN188-207 of the revised manuscript:

“Hydrothermal fluids are typically rich in ammonium (and methane) when circulating through sediment-covered oceanic ridges with modern hydrothermal fluids exhibiting elevated concentrations up to 16 mM^{45–47}. In the Neoproterozoic, some hydrothermal vent fluids also had sufficiently high ammonium concentrations to facilitate the partial N utilisation by abiotic or biotic processes, such as those in the 2.7 Ga Abitibi basin²¹. Ammonium may also be sourced via the remineralisation of organic matter in marine sediments in the absence of hydrothermal fluid flow as, for instance, in the modern Black Sea. However, this process is inconsistent with our N isotope data because nutrient-rich reservoirs that accumulate due to intense basin stratification imply a limited upwelling of nutrients to the surface⁴⁸. In contrast, buoyant hydrothermal vent plumes, which are typically warmer and less dense than the surrounding seawater, enable limited nutrients such as Fe to reach surface waters and stimulate primary productivity in the modern oceans on timescales of ~100 yr^{49,50}. A similar scenario could therefore be plausible for hydrothermal ammonium in the Archean. As ammonium may have been a limiting nutrient in the Neoproterozoic Ocean, it was probably rapidly scavenged from the water column, preferentially removing ¹⁴N via ammonium assimilation and archiving negative N isotope signature in the deep-water shales¹. The remaining pool of ¹⁵N-rich hydrothermal ammonium could then have reached the shallow marine environment via upwelling and fuelled surface biological productivity in the form of microbial communities associated with stromatolites under weakly oxidising conditions due to oxygenic photosynthesis by cyanobacteria (Fig. 5).”

RC6 The implication for the GOE is not convincing. The tectonic backgrounds between these two periods might be substantially different. It seems the GOE was preceded by a tectonic quiescence.

We agree that this aspect of the manuscript could be strengthened. Moreover, in light of the recent publication by Pellertin et al., 2024 (Nature), which was recently published in September 2024, we have shifted the focus of our discussion to the Nitrogen Isotope Event (NIE), to which our samples are directly related. This is still significant with respect to the GOE, because the NIE may be linked to changes in the evolution of biological mechanisms that may have played a role in shaping GOE. This is now stated in LN207-218:

“Overall, the occurrence of strongly positive $\delta^{15}\text{N}$ in the ca. 2.75 Ga Manjeri Fm and 2.73 Ga Cheshire Fm in the Zimbabwe craton further supports the global nature of the NIE, which includes the 2.68 Ga Serra Sul Fm (Amazonian craton) and 2.72 Ga Tumbiana Fm (Pilbara craton). Furthermore, we show that the NIE began at least some 30 Ma

earlier at around 2.75 Ga (Fig. 6), which temporally correlates with globally enhanced volcanism caused by increased magmatic production associated with mantle overturning^{14,16,17}. Therefore, hydrothermally derived fluxes of not only recycled ammonium but also methane and dissolved P (ref. 51) could have triggered biological productivity during this time. In addition, enhanced fluxes of biologically useful transition metals such as copper, molybdenum and zinc^{21,52} may have simultaneously catalysed biological innovations to take advantage of widespread nutrient availability, thereby triggering the necessary conditions for the onset of the NIE and the evolution of life in the buildup to the GOE.”

Minor points:

RC7 Line 23: Is this the first in situ NanoSIMS d15N data report?

These data have been removed following reviewer suggestions and this comment is no longer relevant.

RC8 Line 54: change fractionations to variation range.

We have now reworded this sentence for clarity and is given as the following in LN30-32 of the revised MS:

“The remarkable variation of nitrogen (N) isotope values ($\delta^{15}\text{N} = [(^{15}\text{N}/^{14}\text{N}_{\text{sample}})/(^{15}\text{N}/^{14}\text{N}_{\text{air}})-1] \times 1000$) in Neoproterozoic sedimentary rocks, from -11‰ ¹ up to 50‰ ^{2,3}, hints at fundamental shifts in global marine N cycling prior to the initial rise of atmospheric oxygen during the Great Oxidation Event (GOE).”

RC9 Line 113: -39.1‰? Double check.

We have checked this value and it appears to be correct. We also added a reference to the data source (Table S3).

RC10 Lines 142-144: Double-check the writing here.

This sentence has been completely rewritten and is now (LN129-132):

“However, this mechanism requires the availability of free oxygen in the water column and the local redox conditions must be constrained to determine whether anaerobic²⁷ or aerobic ammonium oxidation⁴ may have occurred.”

RC11 Line 192-195: The molecular tree analysis generally has a large error bar.

We agree that this sentence was speculative and it is now removed in the revised MS.

RC12 Line 247: Delete “d13Corg = “.

We thank the reviewer for noticing this mistake and it is now deleted in the revised MS.

RC13 Line 234 and 235: These two titles have to be changed.

These two sub-sections were related and thus are now merged into one entitled “Stable isotope analyses” (Section 3.1).

END OF REVIEWER 1 COMMENTS

Reviewer #2 (Remarks to the Author):

General comments

RC14 This paper would be timely to publish as it reports another new occurrence of highly enriched $d^{15}N$ values in sedimentary rocks from the 2.6-2.8 Ga time interval. Until very recently such values were so rare that it was not possible to see them as informative about processes shaping the earth system. A new occurrence of these values has just been reported in Pellerin et al 2024 (that to be honest I co-authored) who suggest that they are actually informing about paleo-environmental conditions at the global scale (i.e. crossing the redox threshold for ammonium oxidation), and proposed to name the time period within which these values occur, the Nitrogen Isotope Event (NIE). The present manuscript, by reporting another occurrence of such values comforts the Pellerin et al 2024 proposition of a global event (NIE) and my advice would be to use it to give more weight when making the point of a global process.

We thank the reviewer for their positive feedback. As we have mentioned in previous responses, this paper by Pellerin et al (2024) was just published in Nature in September, a few months after we originally submitted our manuscript. As the reviewer suggests, their findings make it indeed very timely to publish our work. We now cite this new reference accordingly and as suggested by the reviewer, we emphasise our data in relation to the NIE multiple times throughout the revised MS, for instance, in the abstract, introduction (LN33-35), discussion (LN107, 127) and in the closing statements (LN207-218).

RC15 The present manuscript proposes another interpretation for these values: ammonium assimilation from a large ammonium reservoir, a hypothesis that has never been really supported so far for lack of convincing data. The comparison to extremely positive $d^{15}N$ data to other data available from contemporaneous sedimentary formation in the same basin (with less negative and in cases negative $d^{15}N$ values), provide support for the idea that they result from ammonium assimilation in a dominantly anoxic ocean, possibly accompanied by ammonium partial oxidation (The combination of both processes makes it easier to accommodate the extremely high $d^{15}N$ values). To validate the idea that ammonium could be stabilized on anoxic waters, the authors use Ce anomaly and U isotope ratio as redox tracers. They confirm anoxic conditions. Overall, this reasoning seems good to me and I think it should definitively be shared with the community.

Again, we thank the reviewer for their positive feedback. We also agree that the strongly elevated N isotope values may be in fact a combination of both upwelling and partial ammonium oxidation, as proposed by Pellerin et al. (2024), which is specifically stated in LN128-129 of the revised MS:

“In addition, some degree of aerobic N cycling may have occurred, as recently suggested to explain the NIE⁴.”

Moreover, as the hypothesis by Pellerin et al. (2024) also depends on the local redox conditions, we discuss this accordingly using our Ce and U isotope data in Section 2.2. We conclude this discussion by saying there is limited evidence for free oxygen availability in our study area and we prefer the upwelling hypothesis as an explanation for our data in LN179-184:

“Methanotrophic ammonium oxidation to N₂O yields a large isotope fractionation effect and would result a ¹⁵N-rich pool of residual ammonium⁴⁴. However, this process requires oxygen as an electron acceptor and this contradicts the lack of Ce anomalies in our samples, suggesting that the redox potential of the local environment was lower. On this basis, we prefer upwelling as the primary mechanism to explain ¹⁵N enrichments in our study area.”

RC16 From there, the authors take another step by proposing a process allowing this large ammonium reservoir, i.e. hydrothermalism associated with globally enhanced magmatism. To make this point they use evidence for hydrothermalism reported in other publications and spatial variations in the available d¹⁵N values. This is OK for me up to this point. I am not convinced however by the upscaling or the hydrothermalism source of Nitrogen to the basing scale and beyond (as stated in the title). Following my reasoning, this would require a massive amount of ammonium released in many basins while ammonium release is not so frequent in hydrothermalism today, and an initial source of nitrogen which I to my knowledge can only be of sedimentary origin. The proposed process for the buildup of a large ammonium reservoir would thus require that the globally enhanced magmatism is associated with magma emplacement in organic -rich rocks, for which I do not think that we have evidence. I may be mistaken, and may-be they are already out there. In which case this part of the paper just needs additional work to emphasize and discuss them.

We accept that our explanation was not sufficiently supported and have addressed this in the revised manuscript accordingly. First we have changed our title to:

“Anomalous δ¹⁵N values in the Neoproterozoic associated with hydrothermal ammonium upwelling”

This revised title no longer implies hydrothermal ammonium release on a global scale.

As outlined in our response to other comments from the reviewer, we now relate our data in a global context to the NIE (LN209-211). This still notes the temporal correlation between the onset of the NIE (which our data provides a marker for) and globally enhanced volcanism and magmatic activity. This is an important point for explaining what may have triggered the NIE but also explaining the unusually values reported in the Manjeri Fm shales reported by Yang et al. (2019; Nature Geoscience).

RC17 Another weak point in this paper, from my point of view, is the use of Nano-SIMS d¹⁵N results. Using this method, the authors determine the internal d¹⁵N variability of two samples and their bulk value (which is compatible with

the direct bulk d15N measurement). They also use the whole d15N range to comfort their proposed hypothesis of ammonium assimilation.

RC18 I find it tricky to use and discuss these results in such a short paper. In the present form the manuscript lacks detailed explanations and tests to validate the precision of the results. It is therefore not clear to me (non-specialist of Nano-SIMS analyses) if they are meaningful.

RC19 I think removing them from the paper or from the main text would make it easier to read it and would allow to go straight to the point without distracting the reader with data that are less convincing. On the other hand, I admit that such data are rare and need to be presented to the community at some stage. I would suggest to use them in another paper though.

The NanoSIMS have now been removed from the revised MS due to comments from both reviewers.

RC20 If it is decided to maintain this part, I would need more information/clarification in the method section. Here are my questions:

RC21 - Why is a d13C value given for the synthetic N-doped SiC standard and not d15N? how was the 15N ratio determined in the standard? How precisely is it known? how does this precision translate into the d15N determination?

RC22 - As to the matrix effect, it is specified that it adds an additional uncertainty for the N-isotopic ratios of ~15‰? how is it determined in the present study? To which uncertainty are the 15‰ added? in other words, what is the total uncertainty of SIMS d15N measurements?

RC23 - Given that N/C are poorly reproducible (with a factor of 4), I would expect the d15N to be as well. Are there any relationship between d15N and C/N? that could point toward a stronger matrix effect than expected?

RC24 - would the recalculated bulk d15N from nano-SIMMS data be comparable to the bulk value for samples with a low d15N?

As mentioned in response to the previous comments, the Nano-SIMS data have been removed (as suggested by the reviewer) and these comments (RC20-24) are no longer relevant. However, these comments are incredibly useful and will be used in the preparation of a follow-up MS focussing on the Nano-SIMS data.

RC25 Figure 2: can you indicate what is the legend of the data plotted in the figure 2, C? is one symbol equivalent to one measurement or to the average of several measurement of one sample? What is the error bar indicating? According to the text it corresponds to several analyses of two samples. It would be good to add the recalculated bulk values in the figure.

This comment is no longer relevant as it refers to the Nano-SIMS data which has been removed in response to the previous comment and deleted from Figure 2.

RC26 Line 117 to 122: I do not understand the point that is being made here. And I have the feeling it is even counterproductive?

This comment was referring to the inclusion of modern data stromatolites from Shark Bay. We agree with the reviewer than these data were not necessary and have been removed from the revised MS.

RC27 Line 143: add “using” before Ceruim...

This sentence has been reworded in the revised manuscript and the missing article has been added.

RC28 Hoping this helps

We again would like to thank Prof. Ader for taking the time to provide these highly constructive comments and believe that the revised MS is much improved as a result.

END OF REVIEWER 2 COMMENTS

Response to reviewers (NCOMMS-24-38840A)

A point-by-point response document the changes to the manuscript corresponding to the reviewer comments are outlined in this document. Please find the reviewer comments (RC) below in *bold grey italics* and our author responses (AR) in plain black text. Where necessary, we have quoted the appropriate line numbers (LN) in the revised manuscript (MS) with track changes turned off.

Reviewer #1 (Remarks to the Author):

I am generally satisfied with the authors' responses to my comments and their revisions to the manuscript. I have added some further comments to the PDF document attached here for the authors' consideration.

RC1 Title: Personally, I do not like this title very much. The principal mechanism for the anomalous d15N values observed in this time period was primarily caused by microbial ammonium utilization. For incomplete utilization, the d15N values would be very low, like the deep water records, whereas the utilization of the residual ammonium produces high d15N values. Upwelling is ONLY a process that transfers ammonium to shallow water, not the mechanism accounting for the high d15N values. I would like to suggest the authors consider a revision to the title.

We agree with the reviewer and revised the title to:

“Anomalous $\delta^{15}\text{N}$ values in the Neoproterozoic associated with an abundant supply of hydrothermal ammonium”

RC2 LN38: Marine or lacustrine environment?

We have clarified that this refers to a refer lacustrine setting. In the revised manuscript, this is now (LN 36-38):

“Previously, such occurrences have explained locally in terms of either partial nitrification coupled to denitrification in a marine oxygen oasis³ or NH_3 volatilisation under high-pH conditions in a lacustrine setting²”

RC3 LN145-149: I agree that these two can record primary seawater signals at the beginning. However, it does not mean that these two could experience similar variations during diagenesis. I think it is possible that the U isotope composition is altered, while the Y/Ho ratio is not during diagenesis. Right?

The generally held view regarding the effects of carbonate diagenesis on the U isotope and REE systems is that both of these systems are robust. To support our point, we now specifically state this and reference the following well-cited review paper:

vS Hood, A., Planavsky, N.J., Wallace, M.W. and Wang, X., 2018. The effects of diagenesis on geochemical paleoredox proxies in sedimentary carbonates. *Geochimica et Cosmochimica Acta*, 232, pp.265-287.

This is now written as (LN 150-153):

“Despite Y/Ho and U isotope representing two different chemical systems, they may record signals from the same source, i.e. a primary seawater signal and both U isotope and REY signatures are typically well preserved during carbonate diagenesis³⁴.”

RC4 LN148: More contribution from detrital U?

Yes and we now specifically state this in the revised manuscript (LN 156-157):

“This is consistent with a greater influence of detrital material for carbonates with higher $\delta^{238}\text{U}$ and lower Y/Ho.”

RC5 LN155: The $\delta^{238}\text{U}$ values are very close to the modern seawater. Does it mean that the oceans in these two periods are similar in oxidation state from the perspective of $\delta^{238}\text{U}$?

This is an important point to clarify. We now specifically state that this is not the case in the revised manuscript (173-175):

“Although these $\delta^{238}\text{U}$ values overlap with the average value for modern open seawater (ca. -0.4‰^{36}), we do not propose that the average oxidation state of the Neoproterozoic Ocean was similar to present.”

Moreover, we now clearly highlight the main significance of these values in the revised manuscript (LN160-162):

“The main significance of lower $\delta^{238}\text{U}$ in Neoproterozoic carbonates is the implied presence of oxidised U^{6+} in the water column.”

RC6 LN167-171: The argument here means the marine oxidation state was between $\text{Ce}^{4+}/\text{Ce}^{3+}$ and $\text{U}^{6+}/\text{U}^{4+}$. My question in my previous review is this: could this oxidation state oxidize ammonium?

Ammonium oxidation can occur at a redox potential of around +0.4 V under a circumneutral pH. This is now stated in the revised manuscript (LN 178-182):

“Ammonium oxidation can occur at a redox potential of around +0.4 V under a circumneutral pH^{44} , which lies between the reduction potentials of $\text{Ce}^{4+}/\text{Ce}^{3+}$ and $\text{U}^{6+}/\text{U}^{4+}$. Thus, the strongly elevated $\delta^{15}\text{N}$ suggest that the marine redox environment was at least transiently oxidising enough for ammonium oxidation to occur.”

RC7 LN183: Upwelling is not the fundamental factor.

Upwelling has been replaced with “biological ammonium assimilation” in the revised manuscript. In the revised manuscript, this is now rewritten as (LN 190-193):

“As there is a large isotope fractionation effect associated with methanotrophic ammonium oxidation to N_2O , this may yield elevated $\delta^{15}\text{N}$ values⁴⁷, especially in

combination with the upwelling of a ^{15}N -rich pool of residual ammonium due to biological ammonium assimilation in the deep basin¹.”

END OF REVIEWER 1 COMMENTS

Reviewer #2 (Remarks to the Author; copied from pdf file):

RC8 I think that the comments from the first round of reviews have been taken into account. The proposition of hydrothermalism ammonium supply large enough for allowing isotope expression of ammonium assimilation now seems totally convincing to me. I confirm that at the time of this paper submission, Pellerin et al' paper had not been published. I apologize if I did not make this clear. My intention was not to make an accusation but to allow the paper to take this into account in order to go a step further. I have only minor comments to make:

RC9 Line 17-18 please rephrase to make it apparent that the NIE does not correspond only to the positive $\delta^{15}\text{N}$ values but to the time period when they occur.

We thank the reviewer for pointing out this imprecise language. In the revised manuscript (LN17-18), this is now:

“Unusually high $\delta^{15}\text{N}$ values in the Neoproterozoic sedimentary record in the time period from 2.8 to 2.6 Ga, termed the Nitrogen Isotope Event (NIE)”

RC10 Line 33 similarly, rephrase as “the time interval that contains this particular cluster...”

We have mostly accepted this change but prefer “feature” rather than “contain”. In the revised manuscript, this is now written as (LN 33-34):

“The time interval that features this particular cluster of strongly positive N isotope values around 2.8 to 2.6 billion years ago (Ga)”

RC11 Although in most places it is acknowledged that ammonium partial oxidation may contribute to the $\delta^{15}\text{N}$ signature, it seems to me that lines 130-133 and 183-185 may have been overlooked since they argue for the absence of molecular O_2 , and hence for no or negligible ammonium oxidation.

We have specifically clarified this point (and our assumptions) in LN 126-128:

“The highest $\delta^{15}\text{N}$ values may also be explained by partial ammonium oxidation, as recently suggested to explain the NIE⁴, which implies the availability of free oxygen in the water column.”

And LN 190-193:

“As there is a large isotope fractionation effect associated with methanotrophic ammonium oxidation to N_2O , this may yield elevated $\delta^{15}\text{N}$ values⁴⁷, especially in

combination with the upwelling of a ¹⁵N-rich pool of residual ammonium due to biological ammonium assimilation in the deep basin¹"

RC12 Table 2: If the carbonate content was determined by the mass loss during acid attack as stated in the method section, it should be indicated in this table. This would then make it apparent that the TOC percent reported in Table 2 (which is by definition the organic content of the bulk sample) is in fact the TOC_{decarb} (i.e. the TOC content of the decarbonated fraction). Please either change TOC to TOC_{decarb} or calculate the TOC.

Added decarb subscripts and footnote explanation to Table 2 in the revised manuscript.

RC13 It might be necessary to change this in other places in the manuscript (such as the method section, figure 2b).

Updated Figure 2a and the methods section in the revised manuscript.

RC14 Finally, reading this second version, I wonder if it would be worth mentioning that this ammonium assimilation hypothesis also explains both the extremely high values and the extreme variability of d¹⁵N in the NIE. The Neoproterozoic might indeed be particularly prone to ammonium supply by hydrothermalism globally, but not all places would receive enough ammonium to produce those extremely positive d¹⁵N values. I find that this would give even more weight to the paper but this is just a suggestion.

We thank the reviewer for raising this interesting point and have integrated this into the revised manuscript (LN 222-223).

END OF REVIEWER 2 COMMENTS